# WeakM3D: Towards Weakly Supervised Monocular 3D Object Detection

**Liang Peng**[1,2]**, Senbo Yan**[1,2]**, Boxi Wu**[1]**, Zheng Yang**[2]**, Xiaofei He**[1,2] **& Deng Cai**[1,2]
[1]State Key Lab of CAD&CG, Zhejiang University      [2]FABU Inc.
`{pengliang, yansenbo, wuboxi}@zju.edu.cn`
`{yangzheng}@fabu.ai`
`{xiaofeihe, dengcai}@cad.zju.edu.cn`

## Abstract

Monocular 3D object detection is one of the most challenging tasks in 3D scene understanding. Due to the ill-posed nature of monocular imagery, existing monocular 3D detection methods highly rely on training with the manually annotated 3D box labels on the LiDAR point clouds. This annotation process is very laborious and expensive. To dispense with the reliance on 3D box labels, in this paper we explore the weakly supervised monocular 3D detection. Specifically, we first detect 2D boxes on the image. Then, we adopt the generated 2D boxes to select corresponding RoI LiDAR points as the weak supervision. Eventually, we adopt a network to predict 3D boxes which can tightly align with associated RoI LiDAR points. This network is learned by minimizing our newly-proposed 3D alignment loss between the 3D box estimates and the corresponding RoI LiDAR points. We will illustrate the potential challenges of the above learning problem and resolve these challenges by introducing several effective designs into our method. Codes are available at https://github.com/SPengLiang/WeakM3D.

## 1 Introduction

3D object detection is essential for many applications in the real world, such as robot navigation and autonomous driving. This task aims to detect objects in 3D space, bounding them with oriented 3D boxes. Thanks to a low deployment cost, monocular-based methods Chen et al. (2016); Mousavian et al. (2017); Roddick et al. (2018); Liu et al. (2019); Manhardt et al. (2019); Ma et al. (2020; 2021) are drawing increasing attention in both academia and industry. In recent years, monocular 3D detection has achieved remarkable progress. However, it also requires numerous 3D box labels (locations, dimensions, and orientations of objects) for training. These labels are labeled on LiDAR point clouds, where the manually annotating process is quite time-consuming and expensive. The high annotation cost encourages us to dispense with the reliance on the 3D box annotations.

To this end, in this paper we propose WeakM3D, a novel method towards weakly supervised monocular 3D object detection. Considering the well-developed 2D detection technology Redmon & Farhadi (2018); Ren et al. (2015); He et al. (2017); Qi et al. (2018); Zhou et al. (2019), we use an off-the-shelf 2D detector Qi et al. (2018) to obtain 2D boxes, which are then lifted to 3D boxes by predicting required 3D box parameters. To learn the 3D information desired in the lifting process, we employ the LiDAR point cloud as the weak supervision since it provides rich and accurate 3D points within the scene. Specifically, given a raw LiDAR point cloud, we select LiDAR points if their projections are inside the 2D object on the image plane. We term these points object-LiDAR-points, which describe a part outline of the object in 3D space. Therefore, as shown in Figure 1 (a), object-LiDAR-points can be used for aligning with 3D box predictions in loss functions, consequently endowing the network with the ability of being trained without any 3D box label.

However, how to formulate the loss between 3D box predictions and object-LiDAR-points is considerably challenging. We pose and summarize the main challenges and our solutions.

● **Challenge 1** (Section 3.2): as shown in Figure 1 (a), the alignment gap between 3D box estimates and object-LiDAR-points should be measured appropriately. We address this challenge by introduc-

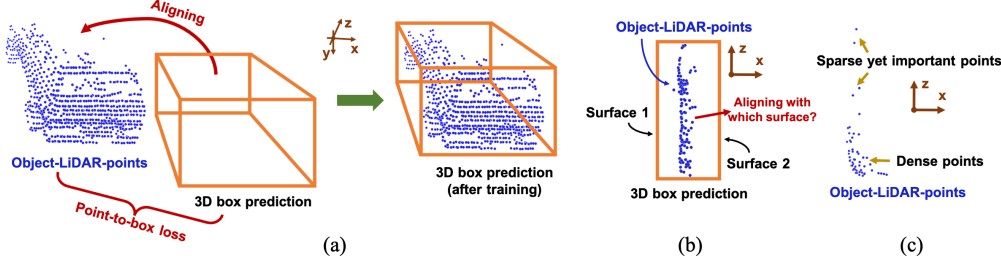

Figure 1: We use the LiDAR point cloud as the weak supervision in training. (a): the aligning process; (b): the alignment ambiguity problem; (c): unevenly distributed LiDAR points.

ing a geometric point-to-box alignment loss, to minimize spatial geometric distances from 3D box predictions to object-LiDAR-points. This loss allows the network to learn the object's 3D location.

• **Challenge 2** (Section 3.3): as shown in Figure 1 (b), an alignment ambiguity problem is caused if only geometrically aligning 3D box predictions with object-LiDAR-points. Regarding object-LiDAR-points that are captured from only one surface of an object, it is not clear which surface of the 3D box prediction should be used for aligning, thus causing an alignment ambiguity issue. Inspired by the constraints of camera imaging and LiDAR scanning, we eliminate this ambiguity by proposing a ray tracing loss. In particular, we track each object-LiDAR-point from the camera optical center and use the resulting ray to make collision detection with predicted 3D boxes to find the surface correspondence. In this way, the alignment ambiguity issue can be alleviated greatly.

• **Challenge 3** (Section 3.4): as shown in Figure 1 (c), LiDAR point clouds distribute in 3D space unevenly. Considering that both geometric alignment and ray tracing losses are calculated point-wisely, unevenly distributed points can cause unbalanced point-wise losses in training, which is harmful since losses produced by sparse yet important points are overwhelmed by losses of dense points. We use the point density to balance point-wise losses to resolve this problem.

• **Challenge 4** (Section 3.5): a 3D box is parameterized by many estimates (locations, dimensions, and orientations). Such entangled 3D box estimates result in a heavy learning burden during training. To resolve this issue in training, we disentangle the learning of each group of estimates by freezing the object dimension and heuristically obtaining the orientation from object-LiDAR-points.

Such challenges and our method are detailed in the following sections. Extensive experiments validate the effectiveness of our method. In summary, our contributions can be listed as follows: **Firstly**, we explore a novel method (WeakM3D) towards weakly supervised monocular 3D detection, removing the reliance on 3D box labels. **Secondly**, we pose the main challenges in WeakM3D and correspondingly introduce four effective strategies to resolve them, including geometric alignment loss, ray tracing loss, loss balancing, and learning disentanglement. **Thirdly**, evaluated on the KITTI benchmark, our method builds a strong baseline for weakly supervised monocular 3D detection, which even outperforms some existing fully supervised methods which use massive 3D box labels.

## 2 RELATED WORK

### 2.1 LIDAR-BASED 3D OBJECT DETECTION

The LiDAR device is able to provide point clouds with precise depth measurements for the scene. Thus, LiDAR-based methods Shi et al. (2019b); Lang et al. (2019); He et al. (2020); Shi et al. (2020); Shi & Rajkumar (2020); Shi et al. (2019a); Zheng et al. (2021) attain high accuracy and can be employed in autonomous driving. Early methods project point clouds into the bird's-eye-view Chen et al. (2017b) or front-view Li et al. (2016), ignoring the nature of point clouds, thus resulting in sub-optimal performances. LiDAR-based 3D detectors can be roughly divided into two categories: voxel-based methods Zhou & Tuzel (2018); Yan et al. (2018); Kuang et al. (2020) and point-based methods Shi et al. (2019a); Qi et al. (2018); Yang et al. (2019). The former partition the 3D space into voxel grids, transforming the irregular raw point cloud to regular voxels so that 3D convolutions can be employed to extract more discriminative features. Point-based methods directly design the network tailored to the raw point cloud representation. Both two types of methods have

achieved great success, but are inherently limited by the main shortcomings of the LiDAR device, *i.e.,* the high price and limited working ranges.

## 2.2 MONOCULAR-BASED 3D OBJECT DETECTION

In recent years, monocular 3D object detection has achieved significant improvements Qin et al. (2019); Brazil & Liu (2019); Ma et al. (2021). Prior works such as Mono3D Chen et al. (2016) and Deep3DBox Mousavian et al. (2017) mainly take advantage of geometry constraints and auxiliary information. More recently, Monodle Ma et al. (2021) resorts to reducing the localization error in monocular 3D detection by three tailored strategies. Furthermore, with the development of depth estimation, some other monocular methods attempt to use the explicit depth information generated from an off-the-shelf depth estimator. Pseudo-LiDAR Wang et al. (2019); Weng & Kitani (2019) converts the image-only representation, to mimic the real LiDAR signal to utilize the existing LiDAR-based 3D detector. PatchNet Ma et al. (2020) rethinks the underlying mechanism of pseudo LiDAR, pointing out the effectiveness comes from the 3D coordinate transform. Although recent monocular 3D object detection methods obtain exciting results, they heavily rely on a large number of manually labeled 3D boxes.

## 2.3 WEAKLY SUPERVISED 3D OBJECT DETECTION

To alleviate heavy annotation costs, some weakly supervised methods are proposed. WS3D Meng et al. (2020) introduces a weakly supervised approach for LiDAR-based 3D object detection, which still requires a small set of weakly annotated scenes and a few precisely labeled object instances. They use a two-stage architecture, where stage one learns to generate cylindrical object proposals under horizontal centers click-annotated in bird's-eye-view, and stage two learns to refine the cylindrical proposals to get 3D boxes and confidence scores. This weakly supervised design does not fully get rid of the dependence on 3D box labels and only works well for LiDAR point clouds input. Another weakly supervised 3D detection method Qin et al. (2020) also takes point clouds as input. It proposes a 3D proposal module and utilizes an off-the-shelf 2D classified network to identify 3D box proposals generated from point clouds. Both two methods cannot be directly applied to fit the single RGB image input. Also, Zakharov et al. Zakharov et al. (2020) propose an autolabeling pipeline. Specifically, they apply a novel differentiable shape renderer to signed distance fields (SDF), leveraged together with normalized object coordinate spaces (NOCS). Their autolabeling pipeline consists of six steps, with a curriculum learning strategy. In contrast to WeakM3D, their method is not end-to-end and rather complicated.

# 3 METHODS

## 3.1 PROBLEM DEFINITION

Given an RGB image captured from a single calibrated camera and the corresponding projection matrix $P$, the task, *i.e.*, monocular 3D object detection, aims to classify and localize objects within the scene in 3D space, where each object is represented by its category $cls$, 2D bounding box $b_{2d}$, and 3D bounding box $b_{3d}$. In particular, we utilize a separate 2D detector to achieve 2D bounding boxes of objects, eliminating the extra attention on this well-developed area. This simple design enables us to focus on solving the 3D bounding box, which is the most important and challenging in monocular 3D detection. Note that we do not use any 3D box label during training. Specifically, the 3D bounding box $b_{3d}$ is parameterized by the location $(x_{3d}, y_{3d}, z_{3d})$, the dimension $(h_{3d}, w_{3d}, l_{3d})$, and the orientation $(\theta_y)$, which are all described at the camera coordinate system in 3D space.

We show our network architecture in Figure 2, and the forward pass is as follows: deep features are firstly extracted from a single RGB image via an encoder. At the same time, the image is fed into an off-the-shelf 2D detector Qi et al. (2018) to generate 2D bounding boxes. The 2D boxes are then used to obtain specified object features for every object from deep features. Finally, we achieve the 3D box $b_{3d}$ by regressing the required 3D box parameters from object features.

**Object-LiDAR-points.** Object-LiDAR-points contain only LiDAR points located on the object surface, indicating much 3D information. Thus they are ideal weakly supervised signals for monocular 3D detection. We achieve object-LiDAR-points from raw LiDAR point clouds and an off-the-shelf

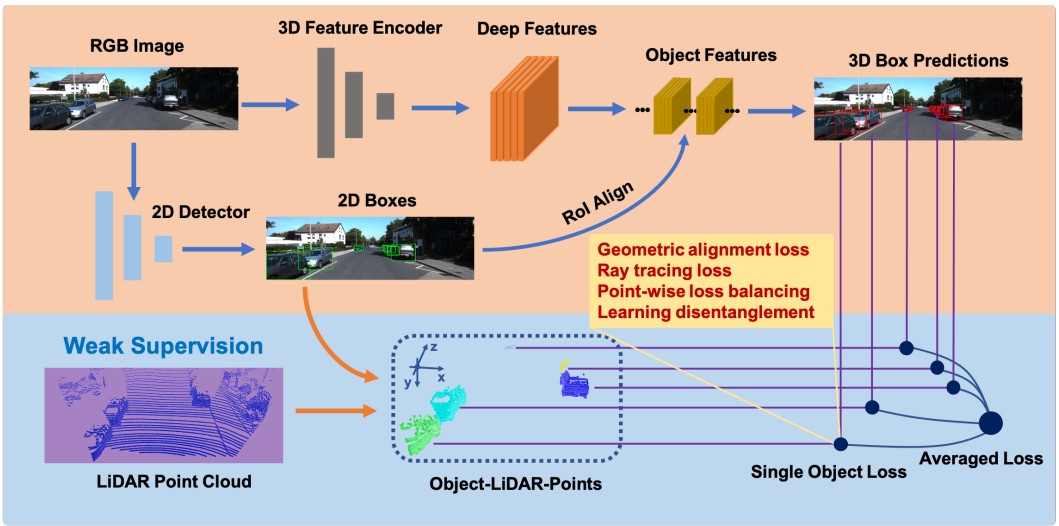

Figure 2: Network architecture. In the inference stage, we require only a single RGB image as input and output corresponding 3D boxes. In the training stage, we use LiDAR point cloud and 2D boxes estimated from a pre-trained model to obtain object-LiDAR-points, which are used to build losses with 3D box predictions to train the network. Best viewed in color.

2D detector. First, the ground plane is estimated from the raw LiDAR point cloud via RANSAC Fischler & Bolles (1981), which is used to remove ground points in the raw point cloud. The remaining LiDAR points whose projections on the image within the 2D box are selected as the initial object point cloud, which contains some points of no interest such as background points. Alternatively, we can also use 2D instance masks from a pre-trained network to select the initial object point cloud. Then, we obtain object-LiDAR-points by filtering the initial point cloud via an unsupervised density clustering algorithm Ester et al. (1996), which is detailed in the appendix.

## 3.2 GEOMETRICALLY ALIGNED 3D BOX PREDICTIONS

Without loss of generality, 3D object boxes should contain object-LiDAR-points and align with them along box edges. To facilitate the learning of the object's 3D location, we impose the location constraint between 3D box predictions from the network and object-LiDAR-points. A naive solution refers to minimizing the euclidean distance from the object center to each LiDAR point. Nevertheless, this trivial loss misleads the network due to its inaccurate nature for localization. As shown in Figure 3, this center distance loss $\mathcal{L}_{center}$ would push the predicted object center to the point cloud as close as possible. Unfortunately, object-LiDAR-points are typically captured from visible surfaces of objects, meaning that the predicted center using only $\mathcal{L}_{center}$ tends to be close to the groundtruth box edge, but not the groundtruth box center.

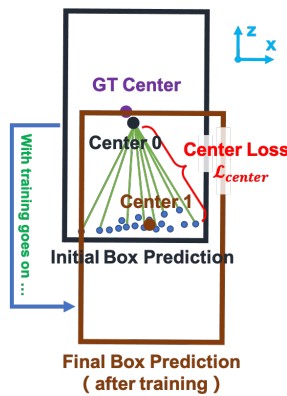

Figure 3: Adverse impacts if using only the center loss. Blue points refer to object-LiDAR-points. Even given an ideal initial 3D box that is close to the groundtruth, the network conducts a much worse prediction after training. Best viewed in color with zoom in.

Based on the above analysis, we propose a geometric alignment loss between the predicted 3D box and associated object-LiDAR-points, in which the major obstacle lies in appropriately measuring the distance from points to the 3D box. Aiming this, we create a ray from the 3D box center $P_{3d}$ to each object-LiDAR-point $P$, where the ray intersects with the edge of the box prediction at $P_I$. Therefore, the geometric alignment loss at each point among object-LiDAR-points is as follows:

$$\mathcal{L}_{geometry} = \|P - P_I\|_1 = \|P - Intersect(Ray_{P_{3d} \to P}, b_{3d})\|_1 \tag{1}$$

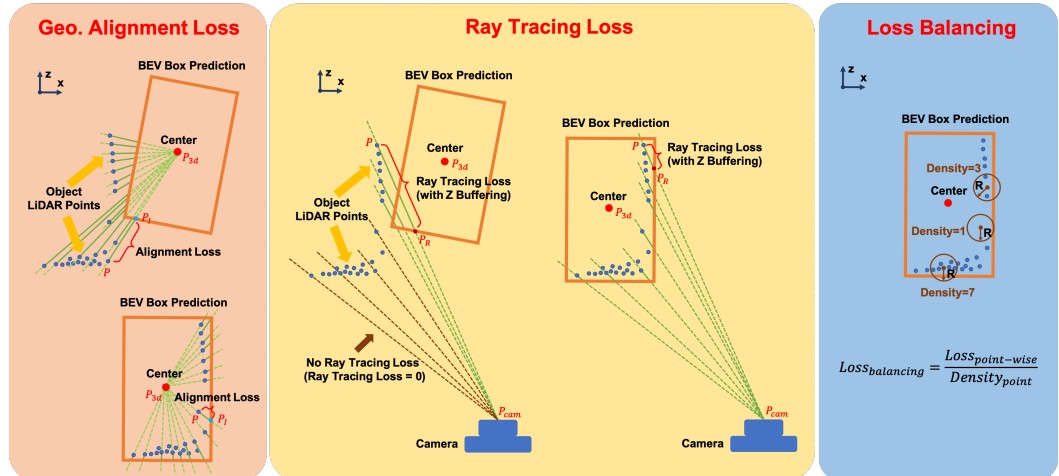

Figure 4: Loss design. We introduce geometric alignment loss (Section 3.2) and ray tracing loss (Section 3.3) with loss balancing (Section 3.4) for each object-LiDAR-point and the corresponding 3D box prediction. The geometric alignment loss focuses on tightly aligning 3D box predictions with object-LiDAR-points. The ray tracing loss further takes occlusion constraints of camera imaging/LiDAR scanning into consideration, to eliminate alignment ambiguity (See Figure 5) in the scene. Best viewed in color with zoom in.

where $Intersect(Ray_{P_{3d} \to P}, b_{3d})$ refers to the intersection point $P_I$. In particular, we do not directly predict the absolute object's 3D center $P_{3d}$, but its projection $[t_x, t_y]$ on the image plane and the object instance depth $z$. Thus, $P_{3d} = [\frac{(t_x - c_x)}{f_x} z, \frac{(t_y - c_y)}{f_y} z, z]^T$, where $f_x, f_y, c_x, c_y$ are the parameters from the camera projection matrix $P$.

We illustrate the geometry alignment loss in Figure 4. This loss forces 3D box predictions to align with object-LiDAR-points in terms of geometry.

### 3.3 RAY TRACING FOR ALIGNMENT AMBIGUITY ELIMINATING

Although the geometric alignment bounds 3D box predictions and object-LiDAR-points, The ambiguity happens when object-LiDAR-points fail to represent an adequate 3D outline of the object, *e.g.*, object-LiDAR-points are captured from only one surface of an object. Specifically, the ambiguity refers to how to semantically decide the correspondence between each object-LiDAR-point and edges of the box prediction in the aligning process. We call this problem the alignment ambiguity.

Figure 5 shows an example. Both the BEV box 0 and 1 commendably align with object-LiDAR-points by the edge 4 and 2, respectively. They conduct the same geometric alignment losses, while their 3D locations are quite different. However, we cannot decide which box is a better prediction if only concerning the geometric alignment because existing geometric clues cannot indicate the semantic correspondence of object-LiDAR-points. In other words, there is a dilemma in choosing the correspondence between box edges and object-LiDAR-points. This ambiguity brings adverse impacts in training, especially for hard samples (their associated object-LiDAR-points usually carry poor information).

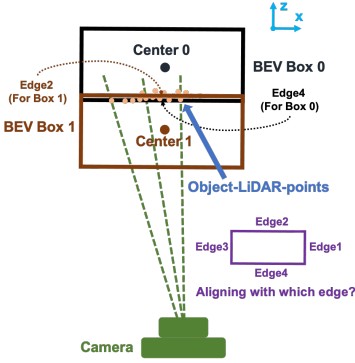

Figure 5: Alignment ambiguity when aligning with object-LiDAR-points. Both box 0 and 1 produce the same geometric alignment losses. Best viewed in color with zoom in.

Generally, we resolve the alignment ambiguity by considering occlusion constraints. Similar to the process of camera imaging, when scanning a scene, the signal of a LiDAR device will be reflected if meeting obstacles. Considering the reflected LiDAR signal inside the camera FOV (field of view), we propose to implement ray tracing from the camera optical center $P_{cam}$ to each object-LiDAR-point, minimizing the distance from each object-LiDAR-point to the intersection point on the object box. As shown in Figure 4, the LiDAR point is $P$ and the intersection point is $P_R$ (with Z buffering, namely, the closer intersection point is chosen), the ray tracing loss is as follows:

$$\mathcal{L}_{ray-tracing} = \begin{cases} \|P - P_R\|_1 & if \ Ray_{P_{cam} \to P} \ intersects \ with \ b_{3d}, \\ 0 & otherwise. \end{cases} \quad (2)$$

$\{P_1, P_2\} = Intersect(Ray_{P_{cam} \to P}, b_{3d})$, and $P_R = P_1$ if $P_1$ is closer to the camera, or $P_2$ otherwise. Note that the ray tracing loss is zero if the ray does not intersect with the predicted 3D box, meaning that this loss here does not contribute to the gradient descent in back-propagation. In this way, we eliminate the alignment ambiguity, encouraging 3D box predictions to follow occlusion constraints in parallel to geometrically aligning with object-LiDAR-points. Let us back to the example shown in Figure 5. The ray tracing losses produced by box 1 are much larger than losses of box 0, consequently leading the network to conduct a reasonable and precise result.

### 3.4 Point-wise Loss Balancing

The varied spatial distribution of object-LiDAR-points is also an obstacle, *i.e.*, the point density is somewhere high while somewhere low. Point-wise losses such as the geometric alignment loss and ray tracing loss all suffer from the unevenly distributed nature. Specifically, the loss produced by the dense region can dominate the total loss, ignoring the loss conducted by other relatively sparse yet essential points. To balance the influence, we normalize the density of object-LiDAR-points when calculating loss. Let us calculate the number of LiDAR points $\mathcal{E}_i$ in the neighborhood at point $P_i$ as follows:

$$\mathcal{E}_i = \sum_j^M \mathbb{1}(\|P_i - P_j\|_2 < R) \quad (3)$$

where $M$ is the number of object-LiDAR-points. From Equation 3, we can know that points within the spatial range $R$ towards the point $P_i$ are counted for the density. Thus we balance point-wise losses by weighting each point with the density, as follows:

$$\mathcal{L}_{balancing} = \frac{1}{M} \sum_i^M (\frac{\mathcal{L}_{geometry_i} + \mathcal{L}_{ray-tracing_i} + \lambda \mathcal{L}_{center_i}}{\mathcal{E}_i}) \quad (4)$$

We illustrate an example in Figure 4. The balanced loss alleviates the adverse impact brought by the unevenly distributed LiDAR point clouds. Here we employ the center loss for regularization and empirically set $\lambda$ to 0.1, and set the coefficients of geometric alignment and ray tracing loss to 1 by default.

### 3.5 Learning Disentanglement for 3D Box Estimates

As mentioned in Simonelli et al. (2019), complicated interactions of 3D box estimates take a heavy learning burden during training. Prior fully supervised methods can easily deal with this issue by disentangling the learning of each group of 3D box estimates, *i.e.*, imposing losses between each group of predictions and corresponding labels, respectively. However, it is not trivial for the weakly supervised manner due to the absence of 3D box labels. To disentangle the learning, we extract object orientations from LiDAR point clouds. In addition, we use a-priori knowledge about dimensions on the specified category and freeze them because objects that belong to the same class have close sizes, *e.g.*, cars. Such information allows us to solve each group of estimates independently, largely reducing the learning burden.

**Orientation Estimated from Object-LiDAR-points.** We propose a simple yet effective method to obtain the global object orientation from object-LiDAR-points. Intuitively, object-LiDAR-points describe a part 3D outline of the object, implicitly indicating the object orientation $\theta_y$. We obtain the object orientation from the directions of paired points. Specifically, as shown in Figure 6, we

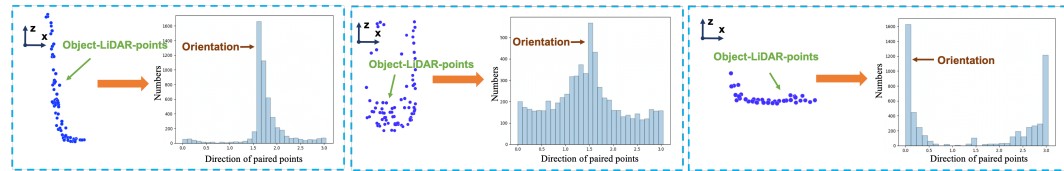

Figure 6: Orientation distribution. We calculate the direction of each pair of points among object-LiDAR-points and draw the histogram. We can observe that the direction that is highest in the histogram is highly related to the object orientation. Best viewed in color with zoom in.

calculate the direction of each pair of object-LiDAR-points, drawing a histogram with respect to the direction range from $0$ to $\pi$. The direction of paired points varies near the object orientation, and the direction $\alpha$ that is highest in histogram refers to $\theta_y$ or $\theta_y \pm \pi/2$. The object orientation is further decided by the spatial offset $d_x$ of object-LiDAR-points along $x$ axis in 3D space. A low offset indicates the orientation close to $\pi/2$ while a high offset denotes the orientation close to $0$. We set the offset threshold to 3.0 meters by default. Therefore, we heuristically obtain the global object orientation, which can be used as the supervised signal for orientation estimates and employed in the process of 3D box alignment. More details can be found in the appendix.

## 3.6 Network Training

We conduct point-wise losses on the bird's-eye-view (BEV) to simplify the task. We can train the network to learn the object's location $(x_{3d}, z_{3d})$ by $\mathcal{L}_{balancing}$, which is the most challenging and crucial. To further obtain the 3D location, we averaging coordinates among object-LiDAR-points along $y$ axis, denoted by $y_L$, building the loss: $\mathcal{L}_{loc_y} = SmoothL_1(\hat{y}, y_L)$, where $\hat{y}$ is the network prediction. Thus, we obtain the overall loss formulation as:

$$\mathcal{L} = \mathcal{L}_{balancing} + \mathcal{L}_{loc_y} + \mathcal{L}_{orient} \tag{5}$$

We empirically set coefficients for each loss item to 1 by default. For the orientation, only the local orientation, *i.e.*, the observation angle, can be directly estimated from the RGB image. Thus the network conducts local orientation estimates and then converts to global orientations using the predicted 3D center $(x_{3d}, z_{3d})$. The orientation loss $L_{orient}$ follows Deep3DBox Mousavian et al. (2017). We provide more details in the appendix.

## 4 Experiments

### 4.1 Implementation Details

Our method is implemented by PyTorch Paszke et al. (2019) and trained on a Titan V GPU. We use the Adam optimizer Kingma & Ba (2014) with an initial learning rate of $10^{-4}$. We train our network for 50 epochs. To obtain an initial object point cloud, we adopt Mask-RCNN He et al. (2017) pre-trained on COCO Lin et al. (2014). Alternatively, we can use an off-the-shelf 2D detector Qi et al. (2018). Both manners lead comparable and satisfactory performance. For the frozen dimensions for cars, we empirically adopt $1.6, 1.8, 4.0$ meters as the height, width, and length, respectively. $R$ for the point density in Equation 3 is set to 0.4-meter. We adjust $y$ coordinate in the location according to 2D-3D consistency Brazil & Liu (2019). We provide more experimental results and discussion in the appendix due to the space limitation.

### 4.2 Dataset and Metrics

Like most prior fully supervised works do, we conduct experiments on KITTI Geiger et al. (2012) dataset. KITTI object dataset provides 7,481 images for training and 7,518 images for testing, where groundtruths for testing are kept secret and inaccessible. Following the common practice Chen et al. (2017a), the 7,481 samples are further divided into training and validation splits, containing 3,712 and 3,769 images, respectively. To utilize raw LiDAR point clouds since our method does not require 3D box labels, we use raw sequences for training that does not overlap with the validation and test

Table 1: The experimental results on KITTI validation set for car category. We can observe that our method even outperforms some fully supervised methods. All the methods are evaluated with metric $AP|_{R_{11}}$, as many prior fully supervised works only provided $AP|_{R_{11}}$ results.

| Approaches | Supervision | $AP_{BEV}/AP_{3D}$ (IoU=0.7)$|_{R_{11}}$ | | |
| --- | --- | --- | --- | --- |
| | | Easy | Moderate | Hard |
| Mono3D Chen et al. (2016) | Full | 5.22/2.53 | 5.19/2.31 | 4.13/2.31 |
| Deep3DBox Mousavian et al. (2017) | | 9.99/5.85 | 7.71/4.10 | 5.30/3.84 |
| OFTNet Roddick et al. (2018) | | 11.06/4.07 | 8.79/3.27 | 8.91/3.29 |
| RoI-10D Manhardt et al. (2019) | | 14.50/10.25 | 9.91/6.39 | 8.73/6.18 |
| MonoDIS Simonelli et al. (2019) | | 24.26/18.05 | 18.43/14.98 | 16.95/13.42 |
| FQNet Liu et al. (2019) | | 9.50/5.98 | 8.02/5.50 | 7.71/4.75 |
| MonoPSR Ku et al. (2019) | | 20.63/12.75 | 18.67/11.48 | 14.45/8.59 |
| M3D-RPN Brazil & Liu (2019) | | 25.94/20.27 | 21.18/17.06 | 17.90/15.21 |
| Pseudo-Lidar Wang et al. (2019) | | 31.88/24.12 | 20.84/15.74 | 18.92/14.96 |
| D4LCN Ding et al. (2020) | | 26.00/19.38 | 20.73/16.00 | 17.46/12.94 |
| RTM3D Li et al. (2020) | | 25.56/20.77 | 22.12/16.86 | 20.91/16.63 |
| PatchNet Ma et al. (2020) | | **32.30/25.76** | 21.25/17.72 | 19.04/15.62 |
| Monodle Ma et al. (2021) | | 30.77/23.29 | **24.53/20.55** | **23.32/17.90** |
| WeakM3D | Weak | 24.89/17.06 | 16.47/11.63 | 14.09/11.17 |
| WeakM3D-PatchNet | | 29.89/20.51 | 18.62/13.67 | 16.06/12.02 |

Table 2: Comparisons on KITTI testing set for car category.

| Approaches | Supervision | $AP_{BEV}/AP_{3D}$ (IoU=0.7)$|_{R_{40}}$ | | |
| --- | --- | --- | --- | --- |
| | | Easy | Moderate | Hard |
| FQNet Liu et al. (2019) | Full | 5.40/2.77 | 3.23/1.51 | 2.46/1.01 |
| ROI-10D Manhardt et al. (2019) | | 9.78/4.32 | 4.91/2.02 | 3.74/1.46 |
| **WeakM3D** | Weak | **11.82/5.03** | **5.66/2.26** | **4.08/1.63** |

set. We do not perform data augmentation during the training. Also, prior fully supervised mainly evaluate their methods on the category of car, and we follow this line. For evaluation metrics, $AP_{11}$ is commonly applied in many prior works, while $AP_{40}$ Simonelli et al. (2019) is suggested to use recently. Most fully supervised methods use IOU 0.7 and weakly supervised methods use IOU 0.5 criterion, thus we follow them for comparisons, respectively.

## 4.3 COMPARING WITH OTHER APPROACHES

**Comparing with monocular-based fully supervised methods.** As shown in Table 1, although our method does not use any 3D box annotation, it outperforms some prior fully supervised methods. For instance, WeakM3D outperforms FQNet Liu et al. (2019) by **8.45** $AP_{BEV}$ under the moderate setting. Table 2 also shows that our method outperforms some prior SOTA fully supervised methods. We can also observe that there still remains a considerable gap between WeakM3D and current SOTA fully supervised methods Ma et al. (2021), we have faith in greatly

Table 3: Comparisons on CenterNet for car category. We apply our method to CenterNet for comparisons.

| Supervision | $AP_{BEV}/AP_{3D}$ (IoU=0.7)$|_{R_{40}}$ | | |
| --- | --- | --- | --- |
| | Easy | Moderate | Hard |
| Full | 3.47/0.60 | 3.31/0.66 | 3.21/**0.77** |
| **Weak (Ours)** | **5.27/1.23** | **3.99/0.79** | **3.36**/0.71 |

pushing the performance of weakly supervised methods in the future due to the rapid development in the community. Furthermore, we apply our method to PatchNet Ma et al. (2020) in Table 1 and CenterNet Zhou et al. (2019) in Table 3 while keeping other settings the same. Interestingly, our weakly supervised method achieves comparable performance compared to the fully supervised manner, demonstrating its effectiveness.

**Comparing with other weakly supervised methods.** We compare our method with other weakly supervised methods. We report the results in Table 4. To make fair comparisons, we employ our baseline network for Autolabels Zakharov et al. (2020). Note that the original VS3D Qin et al.

Table 4: Comparisons of different weakly supervised methods on KITTI validation set for car category.

| Approaches | Input | Supervision | $AP_{BEV}/AP_{3D}$ (IoU=0.5)$\|_{R_{40}}$ | | |
|---|---|---|---|---|---|
| | | | Easy | Moderate | Hard |
| VS3D Qin et al. (2020) | Point cloud | | 31.59/22.62 | 20.59/14.43 | 16.28/10.91 |
| Autolabels Zakharov et al. (2020) | Single image | Weak | 50.51/38.31 | 30.97/19.90 | 23.72/14.83 |
| **Ours** | Single image | | **58.20/50.16** | **38.02/29.94** | **30.17/23.11** |

Table 5: Ablation study. "Disten." in the table refers to the learning disentanglement, and "Geo. alignment" is the geometric alignment.

| Disten. | Geo. alignment | Ray tracing | Loss balancing | $AP_{BEV}/AP_{3D}$ (IoU=0.5)$\|_{R_{40}}$ | | |
|---|---|---|---|---|---|---|
| | | | | Easy | Moderate | Hard |
| | | | | 0.11/0.08 | 0.16/0.10 | 0.15/0.04 |
| $\checkmark$ | | | | 16.16/11.55 | 11.98/8.16 | 9.21/6.04 |
| $\checkmark$ | $\checkmark$ | | | 51.36/45.04 | 33.08/26.34 | 26.42/21.19 |
| $\checkmark$ | $\checkmark$ | $\checkmark$ | | 55.31/48.35 | 36.28/28.48 | 29.03/22.18 |
| $\checkmark$ | $\checkmark$ | $\checkmark$ | $\checkmark$ | **58.20/50.16** | **38.02/29.94** | **30.17/23.11** |

(2020) takes advantage of the per-trained network Dorn Fu et al. (2018) to estimate image-based depth maps. As mentioned in Simonelli et al. (2020); Peng et al. (2021), Dorn utilizes the leaked data on the KITTI 3D object dataset. Therefore we use the fixed split Peng et al. (2021) and keep the number of LiDAR point clouds the same as our method for fairness. We can observe that our method performs the best, validating its effectiveness.

## 4.4 ABLATION STUDY

To investigate the impact of each component in our method, we conduct extensive ablation studies. As shown in Table 5, the accuracy of the baseline is very low, indicating that the weakly supervised manner is challeng-

Table 6: Comparisons of different weak supervisions in training.

| Weak supervisions | $AP_{BEV}/AP_{3D}$ (IoU=0.5)$\|_{R_{40}}$ | | |
|---|---|---|---|
| | Easy | Moderate | Hard |
| 2D box + LiDAR | 55.94/47.52 | 34.11/26.46 | 25.56/19.88 |
| 2D mask + LiDAR | **58.20/50.16** | **38.02/29.94** | **30.17/23.11** |

ing. By adding the learning disentanglement and geometry alignment between 3D box predictions and object-LiDAR-points, we obtain significant progress. When employing ray tracing, which considerably eliminates the alignment ambiguity, the performance is further improved. Finally, the balancing at point-wise losses in favor of accuracy and stability makes the network perform the best. These results validate the power of our method. We also compare the weakly supervised manner using different requirements in Table 6. We can see that both requirements lead to good performance.

## 5 LIMITATIONS AND FUTURE WORK

There are also some limitations in our work. The loss for 3D localization is a little complicated, and the accuracy can be further improved by a more elegant design. As an emerging area, there is ample room for improving the accuracy of weakly supervised monocular 3D object detection. Therefore, we resort to alleviating the label reliance further and boosting the performance in future work.

## 6 CONCLUSIONS

In this paper, we explore a novel weakly supervised monocular 3D object detection method. Specifically, we take advantage of 2D boxes/masks to segment LiDAR point clouds to obtain object-LiDAR-points, which are employed as the weakly supervised signal for monocular 3D object detection. To learn the object's 3D location, we impose alignment constraints between such points and 3D box predictions. Aiming this, we point out the main challenges of the weakly supervised manner, introducing several novel strategies to deal with them. Extensive experiments show that our method builds a strong baseline for weakly supervised monocular 3D detection, which even outperforms some prior fully supervised methods.

ACKNOWLEDGMENTS

This work was supported in part by The National Key Research and Development Program of China (Grant Nos: 2018AAA0101400), in part by The National Nature Science Foundation of China (Grant Nos: 62036009, U1909203, 61936006, 62133013), in part by Innovation Capability Support Program of Shaanxi (Program No. 2021TD-05).

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

# Appendix

Due to the space limitation, we provide details omitted in the main text in this appendix, which is organized as follows:

## A  NETWORK AND RUNNING TIME

For the feature encoder, we adopt different backbones including ResNet18, ResNet34, and ResNet50 He et al. (2016). We train our network with a batch size of 8 by default. Specifically, we remove the output layer and the last convolution stage of ResNet, and obtain $7 \times 7$ RoI Aligned He et al. (2017) object features from extracted deep features. As shown in Figure 7, we reshape each output tensor to a 1-dim tensor, passing fully connected layers to regress corresponding outputs. We show the quantitative results on different backbones in Table 7. Under the $AP_{BEV}$/$AP_{3D}$ (IoU=0.3) metric, different backbones all show satisfactory and comparable accuracy, demonstrating the robustness of our method. If using the stricter $AP_{BEV}$/$AP_{3D}$ (IoU=0.5) metric, a better backbone is more beneficial for the overall performance as it encodes richer features for object 3D localization.

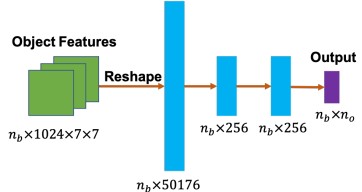

Figure 7: Regression network. $n_b$ denotes the number of objects, and $n_o$ denotes the number of required 3D box parameters.

Table 7 shows the running time for the forward pass at different backbones. Thanks to our lightweight design, the time-cost is little. The adopted 2D detector takes extra 60ms in inference, meaning that the overall running time of our method is less than 80ms, meaning that our method is efficient.

Table 7: The experimental results on different backbones on KITTI validation set.

| Backbone | Time (ms) | $AP_{BEV}$/$AP_{3D}$ (IoU=0.3)$\mid_{R_{40}}$ | | | $AP_{BEV}$/$AP_{3D}$ (IoU=0.5)$\mid_{R_{40}}$ | | |
|---|---|---|---|---|---|---|---|
| | | Easy | Moderate | Hard | Easy | Moderate | Hard |
| ResNet18 | **5.90** | 75.53/70.87 | 56.45/51.20 | 46.83/42.33 | 50.92/44.78 | 33.17/25.57 | 26.23/20.41 |
| ResNet34 | 8.77 | 80.92/77.61 | 59.55/55.10 | 48.69/44.85 | 57.15/48.39 | 36.00/27.14 | 27.90/21.53 |
| ResNet50 | 11.97 | **81.17/78.44** | **59.87/56.42** | **48.98/45.81** | **58.20/50.16** | **38.02/29.94** | **30.17/23.11** |

## B  DETAILED ABLATION STUDIES

We conduct extensive experiments to validate the effectiveness of each component in WeakM3D. As shown in Table 8, the baseline (Experiment a) using only the center loss shows an extremely low

Table 8: Detailed ablation study. "Disten." in the table refers to the learning disentanglement, and "Geo. align." is the geometric alignment.

| Exp. | Disten. | Geo. align. | Ray tracing | Loss balancing | $\text{AP}_{BEV}/\text{AP}_{3D}$ (IoU=0.5)$\|_{R_{40}}$ | | |
| --- | --- | --- | --- | --- | --- | --- | --- |
| | | | | | Easy | Moderate | Hard |
| a | | | | | 0.11/0.08 | 0.16/0.10 | 0.15/0.04 |
| b | √ | | | | 16.16/11.55 | 11.98/8.16 | 9.21/6.04 |
| c | | √ | | | 49.09/37.79 | 24.86/17.73 | 18.24/12.60 |
| d | | | √ | | 32.39/24.79 | 14.56/10.34 | 10.67/7.38 |
| e | | | | √ | 0.77/0.26 | 0.45/0.13 | 0.28/0.11 |
| f | √ | √ | | | 51.36/45.04 | 33.08/26.34 | 26.42/21.19 |
| g | √ | | √ | | 49.92/41.91 | 32.90/25.24 | 27.32/20.50 |
| h | √ | | | √ | 17.79/12.75 | 12.04/7.38 | 9.03/5.60 |
| i | | √ | √ | | 53.02/44.12 | 34.53/27.07 | 26.96/21.90 |
| j | | √ | | √ | 51.56/44.11 | 33.77/27.05 | 26.73/21.10 |
| k | | | √ | √ | 51.44/43.79 | 32.07/26.29 | 25.86/20.69 |
| l | √ | √ | √ | | 55.31/48.35 | 36.28/28.48 | 29.03/22.18 |
| m | √ | √ | | √ | 52.50/47.58 | 33.45/27.93 | 26.31/21.60 |
| n | √ | | √ | √ | 56.25/49.78 | 34.34/27.30 | 26.82/20.76 |
| o | | √ | √ | √ | 56.60/49.03 | 37.37/29.43 | 28.68/21.77 |
| p | √ | √ | √ | √ | **58.20/50.16** | **38.02/29.94** | **30.17/23.11** |

detection accuracy, indicating the difficulty for the weakly supervised manner. When only employing one of the components, we can observe that the geometric alignment (Experiment c) brings the most improvements, as it enables the network to learn the most 3D information. When employing two of the components, we can know that the combination of geometric alignment and ray tracing (Experiment i) performs the best. Such two strategies mutually reinforce each other, allowing the network to effectively learn the object's 3D location. Also, after applying loss balancing (Experiment o), unevenly distributed point-wise losses can be balanced to facilitate the learning process. Finally, if we use all proposed strategies (Experiment p), we can achieve the best performance. Furthermore, instead of freezing the object dimension, we let the network learn the dimension and show the results in Table 9. We can see that adding more entangled learning objectives brings a heavier learning burden, thus making the network perform slightly worse. Our detailed ablation studies comprehensively show the power of our method and how such strategies interact.

Table 9: Ablation study for dimension.

| Dimension | $\text{AP}_{BEV}/\text{AP}_{3D}$ (IoU=0.5)$\|_{R_{40}}$ | | |
| --- | --- | --- | --- |
| | Easy | Moderate | Hard |
| Learned | 56.24/48.37 | 37.04/29.41 | 29.45/22.74 |
| Frozen | **58.20/50.16** | **38.02/29.94** | **30.17/23.11** |

## C  PERFORMANCE ON WAYMO

We also evaluate our method on Waymo Sun et al. (2020) dataset, which is a considerably large dataset, and provide the results in Table 10. Waymo dataset provides many annotations, thus the fully supervised manner method M3D-RPN Brazil & Liu (2019) is supposed to perform much better than our method. However, we can observe that our method obtains comparable performance compared to M3D-RPN. Especially for medium and far objects, our method outperforms it with a significant margin. It indicates that our weakly supervised manner is more suitable on massive data. Also,

Table 10: Comparisons on Waymo. We compare our method with the fully supervised method M3D-RPN. We obtain comparable performance.

| Difficulty | Approaches | AP$_{BEV}$ (IOU=0.5) | | | | AP$_{3D}$ (IOU=0.5) | | | |
|---|---|---|---|---|---|---|---|---|---|
| | | Overall | 0−30m | 30−50m | 50m−∞ | Overall | 0−30m | 30−50m | 50m−∞ |
| LEVEL 1 | M3D-RPN | 8.01 | **24.04** | 3.92 | 0.40 | **5.87** | **19.22** | 2.40 | 0.19 |
| | WeakM3D | **8.72** | 20.39 | **7.73** | **1.14** | 4.81 | 12.20 | **3.78** | **0.46** |
| LEVEL 2 | M3D-RPN | 7.51 | **23.96** | 3.81 | 0.35 | **5.50** | **19.15** | 2.33 | 0.17 |
| | WeakM3D | **8.18** | 20.31 | **7.50** | **0.99** | 4.50 | 12.16 | **3.67** | **0.40** |

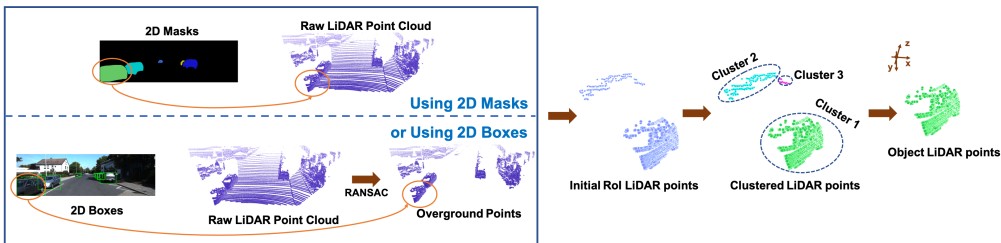

Figure 8: Details of obtaining Object-LiDAR-points. Best viewed in color.

prior point-cloud-based weakly supervised methods and many fully supervised monocular methods do not evaluate their results on Waymo, thus our method sets a good baseline for future works.

Notably, in this comparison, the performance of WeakM3D is still limited by the number of training samples. Such samples can be easily collected in real applications, as we do not require the extra manual annotating process. In other words, our method is inherently suitable for the autonomous driving scenario, which automatically produces more and more unlabeled data, therefore it has much potential to be explored.

## D GENERALIZATION ABILITY

Our method is robust and works well for most object categories. We follow a basic assumption adopted by almost all 3D detection works, *i.e.*, all objects in 3d detection are represented as cuboids. Please consider the underlying mechanism of human annotating. People annotate each object by finding a reasonable minimum 3d bounding box. Correspondingly, our loss functions exactly aim to learn this reasonable minimum 3D bounding box for the object. Our method therefore works well for most object types. To demonstrate this, we conduct experiments on pedestrian and cyclist categories in Table 11, and we obtain promising results. To the best of our knowledge, we are the first weakly supervised method that works for the two classes.

Table 11: Performance on other categories.

| Categories | AP$_{BEV}$(IoU=0.25)$|_{R_{40}}$ | | |
|---|---|---|---|
| | Easy | Moderate | Hard |
| Pedestrain | 3.79 | 3.21 | 3.12 |
| Cyclist | 5.16 | 3.60 | 3.33 |

## E DETAILS OF OBJECT-LIDAR-POINTS

As mentioned in the main text, we can obtain an initial object point cloud by using the raw LiDAR point cloud and 2D boxes/masks. This initial point cloud contains some noisy or background points, which should be removed. As shown in Figure 8, we cluster the initial object point cloud using an unsupervised clustering algorithm Ester et al. (1996), in which points are divided into different clusters in terms of the density. Therefore, we select the cluster with most points as object-LiDAR-points and remove other points. In particular, we further remove the points if their coordinates $y_{3d}$ are lower than the median of object-LiDAR-points. These inner points from the bird's-eye-view perspective do not contribute to the object's 3D location. Then, we randomly sample 100 object-

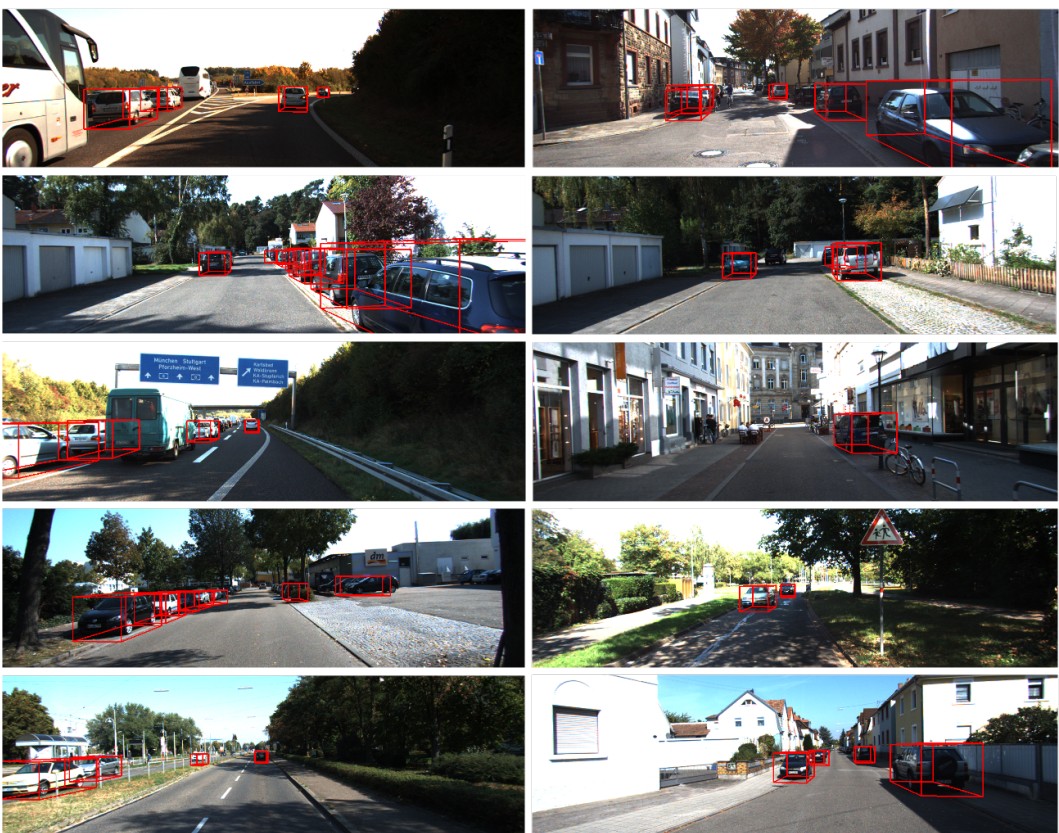

Figure 9: Qualitative results. Best viewed in color.

LiDAR-points. Such points precisely describe a part outline of an object in 3D space, and they are ideal weak supervisions for corresponding 3D box prediction.

## F    QUALITATIVE RESULTS

We provide some qualitative results in KITTI validation set in Figure 9. We can see that our method works well for most cases, while it can fail for the hard cases, *e.g*, truncated, heavily occluded, and far samples. This inspires us to further improve the performance in future work.

## G    ORIENTATION ANALYSIS

In monocular 3D detection, the orientation consists of two types of orientations, *i.e.*, the global orientation $\theta_y$ and the local orientation $\delta_y$, where the local orientation is also named the observation angle. We show the relationship between $\theta_y$ and $\delta_y$ in Figure 10. For monocular 3D detection, the global orientation $\theta_y$ is hard to reason directly. Even holding the same global orientation $\theta_y$, the appearance of an object shown on the image can change as the viewpoint varies. We give an example in Figure 10. For the location 1, 2, and 3, the mainly captured visible surfaces by the camera are different, consequently

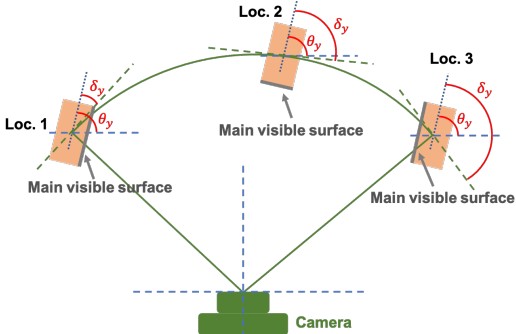

Figure 10: Local orientation $\delta_y$ and global orientation $\theta_y$. The same object with the same global orientation at different viewpoints shows different appearances on the image.

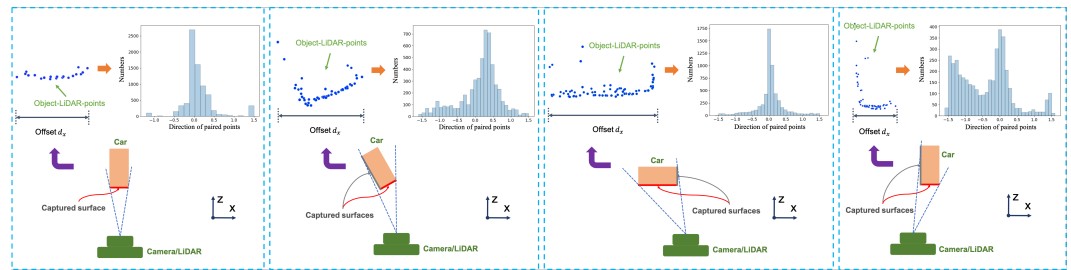

Figure 12: Distribution of directions of paired points. The red line on the box refers to the mainly captured surface, meaning that LiDAR points captured from this surface dominate the object-LiDAR-points. A low offset $d_x$ refers to the orientation close to $\pi/2$ while a high offset indicates the orientation close to $0$ or $\pi$. Best viewed in color with zoom in.

showing varied appearances on the image. In light of this, instead of directly regressing the global orientation $\theta_y$, we estimate the local orientation $\delta_y$ and then convert it to $\theta_y$ by taking the camera viewpoint into account. More details can be found in Mousavian et al. (2017).

Also, we provide details for obtaining the global orientation from Object-LiDAR-points. Object-LiDAR-points denote a part outline of the object, which is highly related to the global orientation $\theta_y$. Specifically, we calculate the direction of each pair of object-LiDAR-points and then draw the corresponding histogram. We show the paired points and the histogram in Figure 11 and Figure 12, respectively. The direction $\alpha_y$ that is highest in histogram refers to $\theta_y$ or the perpendicular direction of $\theta_y$. To further determine the orientation, we use the spatial offset $d_x$ of object-LiDAR-points along $x$ axis in 3D space. Since a car in motion usually faces forward, we first convert $\alpha_y$ to the range $(\frac{\pi}{4}, \frac{3\pi}{4})$ by adding or subtracting $\frac{\pi}{2}$, thus $\theta_y$ is derived as follows:

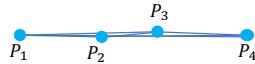

Figure 11: Illustrations of paired points. We connect each pair of points and then calculate the direction of the resulting line.

$$
\theta_y = \begin{cases}
\alpha_y - \frac{\pi}{2} & \text{if } d_x > C \text{ and } \alpha_y \geq \frac{\pi}{2}, \\
\alpha_y + \frac{\pi}{2} & \text{if } d_x > C \text{ and } \alpha_y < \frac{\pi}{2}, \\
\alpha_y & \text{otherwise.}
\end{cases} \tag{6}
$$

where $C$ is the offset threshold. $C$ is set to 3.0 meters by default. From Figure 12 and Equation 6, we can know that a low offset indicates the orientation close to $\pi/2$ while a high offset denotes the orientation close to $0$ or $\pi$.

## H  STATISTICS ON 3D OBJECT LABELS

We conduct statistics on 3D object labels for the car in KITTI Geiger et al. (2012) and show the results in Figure 13. Please note, we do not use the statistics on 3D box labels as the dimension priors. Specifically, we draw the histogram for the object dimension and 3D location. As mentioned in the main text, objects that belong to the same class usually have close sizes. We can observe that the height of cars mainly varies from 1.4 to 1.6 meters. Similarly, the width of cars mainly varies from 1.5 to 1.7 meters, and the width of cars mainly varies from 3.3 to 4.5 meters. In contrast to the low-range varied dimensions, the object's 3D locations vary a lot, especially for $\{x_{3d}, z_{3d}\}$, e.g., the object depth ($z_{3d}$) varies from 0 to 80 meters. Considering the IoU (Intersection over Union) criterion and the ill-posed nature of monocular imagery, improving the localization accuracy is much more important than obtaining more precise object dimensions. The satisfactory performance of our method also indicates that the manner of taking a fixed object dimension is acceptable for weakly supervised monocular 3D detection. Intuitively, in future work we can attempt to obtain more accurate object dimensions to further improve the accuracy.

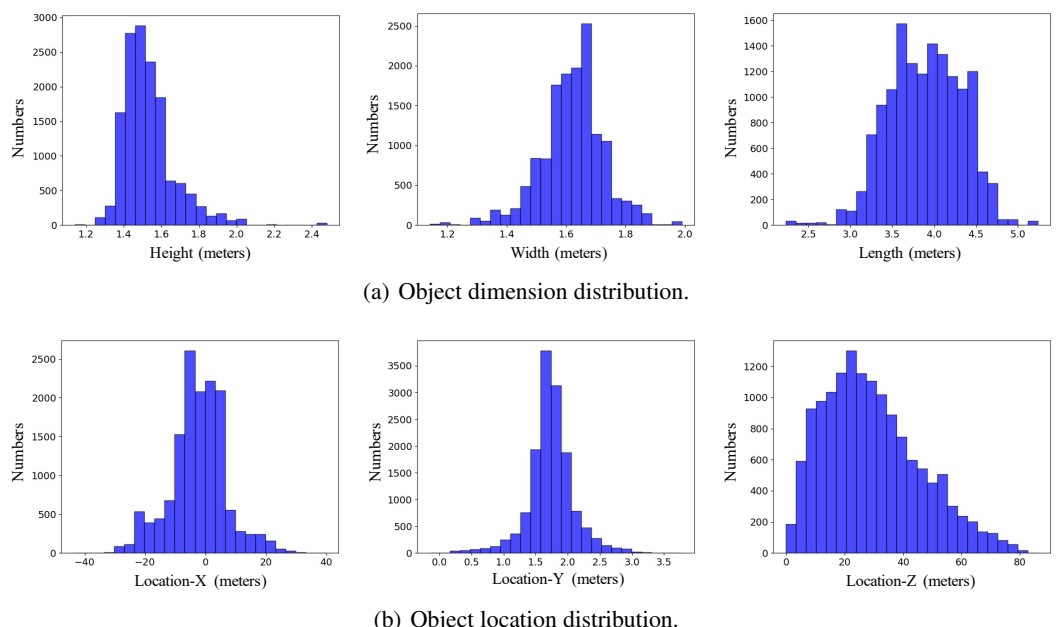

(a) Object dimension distribution.

(b) Object location distribution.

Figure 13: Statistics on 3D object labels on KITTI.

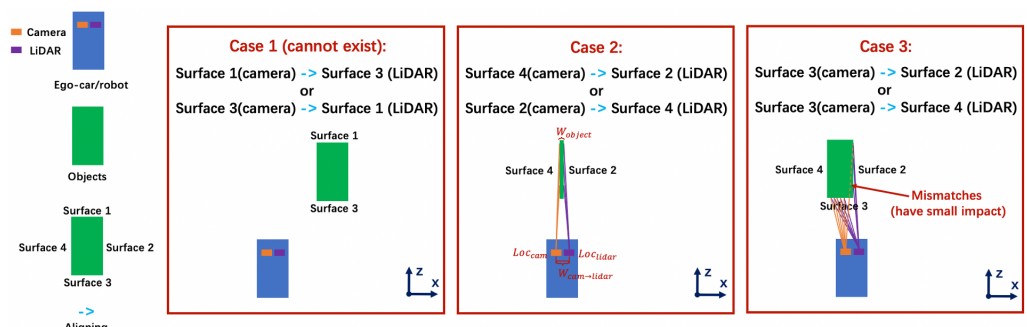

Figure 14: Theoretical failure cases caused by viewpoint differences between the LiDAR and camera. A well known yet important fact is that the LiDAR and camera are installed in the same car/robot, which indicates that their viewpoint differences are marginal compared to the extensive 3D space. We summarize the theoretical failure cases caused by the viewpoint differences. For case 1, it cannot happen since the camera and LiDAR always face towards objects. The occlusion constraints for surface 1 and 3 are the same for the camera and LiDAR. For case 2, the object width $W_{object}$ is lower than $W_{cam->lidar}$ (the offset between camera and LiDAR along $X$ axis) meanwhile the object center is very close to the ego-car/robot center along $X$ axis. Such objects are few. Also, the adverse impact brought by this type of mismatching can be ignored since $W_{object}$ is very small. For case 3, the conditions are similar to case 2, but the object can have a bigger $W_{object}$. In this case, the LiDAR mainly captures points from surface 3, such points dominate the loss since points captured from surface 2 are very few. Thus mismatches on surface 2 can also be ignored in the overall loss. Best viewed in color with zoom in.

## I  DISCUSSION ON VIEWPOINT DIFFERENCES BETWEEN LiDAR AND CAMERA

In the real world, the LiDAR and camera have different viewpoints, which may have some impacts on the proposed losses. Therefore, in this section we discuss the impact brought by viewpoint differences. First, if an object exists in one sensor while invisible in the other sensor, we cannot obtain the associated object-LiDAR-points. In this case, we directly ignore this object. Second, if an object is visible in both camera and LiDAR, the viewpoint differences may bring some impacts in some conditions. For this case, we provide a detailed discussion Figure 14. In sum, we can ignore the impact brought by viewpoint differences.

## J  DISCUSSION ON DIFFICULT CASES

Potential noises and sparsity can exist in clustered points from faraway and small objects. Such faraway and small objects are typical dilemmas for most 3D detection methods due to the inadequate visual clues on the image or few LiDAR points. Unfortunately, our method is a general design and does not introduce special components for such objects. We may use depth completion to enhance the raw LiDAR point cloud, to alleviate this issue. This solution can be explored in future works.

Also, when only one object surface (*e.g.*, only the back of a car) is observed, the network cannot learn the complete object dimensions, and the learning of object locations is also affected by object dimensions. In other words, such LiDAR point clouds only provide part of weak supervision, thus the network trained with them performs suboptimally compared to using more complete LiDAR point clouds. Fortunately, not all LiDAR point clouds are incomplete, and other good LiDAR points provide accurate and complete object 3D information, including object 3D locations and dimensions. Therefore, the network still works well after being trained with different types of LiDAR point clouds.

Table 12: Comparisons on different supervised manners on KITTI validation set for car category when using the our baseline network.

| Supervision | $\text{AP}_{BEV}/\text{AP}_{3D}$ (IoU=0.5)$|_{R_{40}}$ | | |
| | Easy | Moderate | Hard |
|---|---|---|---|
| Full | 53.61/44.78 | 34.84/28.64 | 29.59/**23.90** |
| **Weak (Ours)** | **58.20/50.16** | **38.02/29.94** | **30.17**/23.11 |

Table 13: Comparisons on different supervised manners on KITTI validation set for car category when using PatchNet Ma et al. (2020) baseline.

| Supervision | $\text{AP}_{BEV}/\text{AP}_{3D}$ (IoU=0.5)$|_{R_{40}}$ | | |
| | Easy | Moderate | Hard |
|---|---|---|---|
| Weak (Ours) | 59.41/54.10 | 37.01/32.47 | 28.53/24.27 |
| **Full** Ma et al. (2020) | **64.31/57.94** | **39.45/35.25** | **34.76/30.07** |

## K  DISCUSSION ON DIFFERENT BASELINES

It is interesting to see the performance gap between the fully supervised and our weakly supervised manner on different baselines. For the proposed baseline network in the paper, we report the results in Table 12. Interestingly, on this baseline network, we can see that our method performs better than using the full supervision. This tendency is the same as Table 3, since our method takes more geometry and scene constraints into consideration, benefiting simple baselines. On the other hand, when using more advanced baselines, our method performs worse than using the full supervision (*e.g.*, WeakM3D-PatchNet *vs.* PatchNet in Table 1). It is because advanced baselines weaken the usefulness of geometry and scene constraints of our method. To comprehensively investigate the performance on advanced baselines, we report the results on PatchNet Ma et al. (2020) under the

Table 14: Results on NuScenes validation set on the car category. We provide a baseline for future works.

| Category | AP | ATE | ASE | AAE |
|----------|------|------|------|------|
| Car | 0.214 | 0.814 | 0.234 | 0.682 |

IoU 0.5 criterion in Table 13. We can observe that the advanced baseline can adequately use the full supervision, thus showing better results than our method.

## L    RESULTS ON NUSCENES

To encourage future works on weakly supervised monocular 3D detection, we further provide results on the NuScenes dataset, which can serve as a baseline for future works. We report the results in Table 14. We use the pre-trained Mask-RCNN, to obtain RoI LiDAR points in training, and to obtain 2D detections in inference. Please note that the pre-trained Mask-RCNN cannot output all classes annotated in NuScenes, we only conduct experiments on the car category. Additionally, we do not report the AVE (Average Velocity Error) and AOE (Average Orientation Error) because we can not obtain supervision of the velocity and the moving direction in the weakly supervised method.

