# OpenReview forum: "WeakM3D: Towards Weakly Supervised Monocular 3D Object Detection"
_ICLR.cc/2022/Conference — ICLR 2022 Poster_

### Official Review · Reviewer_D4LN · 2021-10-23

**Correctness:** 4
**Technical Novelty And Significance:** 2
**Empirical Novelty And Significance:** 3
**Recommendation:** 6
**Confidence:** 4

**Main Review:**

The main contribution of the paper is the novel method for the task that has not received much attention but is important. The proposed method is simple and intuitively makes sense for cars in driving scenes that is the most important 3D object category for detection. The paper is well written and the experiments clearly show that each of the proposed components is effective (Table 5).

However, I think the novelty of the task might be limited as an important paper is missing from the reference. The paper is "Autolabeling 3D Objects with Differentiable Rendering of SDF Shape Priors" by Zakharov et al. presented at CVPR 2020. This paper addresses the task of learning a 3D object detector from unlabeled images and point clouds.

The proposed method is different from [Zakharov et al. 2020], but it is unclear which is better. However, it is unclear which is better, because the differences are not discussed and no experimental comparison are provided. The percentage of performance drop (10%~20%) due to the elimination of supervision seems to be almost the same in the two papers, but direct comparison is not possible due to the different baseline methods and evaluation metrics. This submission would be much strengthen if an experiment to compare with [Zakharov et al. 2020] is included. I think discussion on the differences in assumptions and conditions would needs to be included.

I am also concerned that the number of object categories to which the proposed method can be applied may be small. The proposed loss function assumes that point clouds are located near the edge of a bounding box. This assumption holds for a rectangular object such as a car. However, it does not for many other categories such as pedestrians and cyclists, and I suspect that the proposed method would rather worsen detection performance. It is fine that it can only be used for cars, but I think it would be better to clearly state so.

I think the optimal 3D location that minimizes the loss function can be done as a preprocessing step, similar to what is done for rotation angles. I am interested to see how the performance changes with this "autolabeling" setting. I guess it will make training stabler. (This is just an interest, not a request for additional experiments.)

Other concerns

- The results in Table 3 seems strange because the accuracy with weaker supervision is consistently higher than that with full supervision. Maybe the first and second rows are swapped?

**Summary Of The Paper:**

This paper addresses the task of learning monocular 3d object detection from unlabeled images and LiDAR point clouds with the help of off-the-shelf 2D detection and segmentation models. This task is usually performed by predicting the 3D location, 3D size, and 1D rotation of an object in a 2D bounding box (BB). Since no 3DBB labels are provided, the authors propose to fix the size to a predetermined value and use heuristically estimated rotation angles from the point cloud as supervision. For 3D location, the authors propose a novel loss functions. The assumptions are that (1) the points belonging to an object are located near the surface of a BB and (2) the surface of a BB with points is visible to the camera.

The experimental results show that the proposed method can eliminate the need for 3DBB labels at the expense of 10~20% lower performance of existing detectors (PatchNet in Table 1 and CenterNet in Table 3).

The claimed contributions are (1) the novelty of the task, (2) a novel method for it, and (3) its good performance on the KITTI benchmark.

**Summary Of The Review:**

This paper proposes an effective method for an important but under-addressed issue. However, the novelty of the task is limited and the comparison with prior methods does not seem sufficient.

---

### Official Review · Reviewer_Tcr1 · 2021-10-29

**Correctness:** 3
**Technical Novelty And Significance:** 3
**Empirical Novelty And Significance:** 3
**Recommendation:** 6
**Confidence:** 5

**Main Review:**

Strengths:
+ The motivation is good. This method uses object-LiDAR points to supervise the 3D prediction of monocular images. Such practice can solve the problem of high-cost 3D annotation.
+ The writing and organization of this paper are good. The challenges described in the introduction are well-addressed by proposed solutions.
+ The idea is easy to follow. Both the main components and details are clearly explained.
+ Experimental results demonstrate the effectiveness of the proposed method.

Weakness & Questions
- The proposed method has strict requirements on the sensor configuration. The LiDAR and camera need to be well-calibrated and synchronized. Such requirements limit the generality of the method.
- The key component of the weakly supervised training is the selection and filtration of object-LiDAR points. Although the authors use ground-plane subtraction, 2D-box selection, and unsupervised clustering. The quality of object points is still questionable, especially the long-range and small objects. How to relieve the problems of potential noises and sparsity in such clustered points?
- The formulation of Equation 3 is not very accurate. The $\varepsilon$ is the number of LiDAR points in the neighborhood but not the density. The lower part of the summary operator is suggested to add $j!=i$.
- In the learning disentanglement, the dimensions of objects are directly indicated by prior knowledge. However, the sizes of objects in the same category are very different. For example, the length of the vehicle varies from 3m to 5m. Is that reasonable to use a single value to represent the whole category? How about the average scale error (ASE)[1] of final detection?
- The authors report the running time for the forward pass in the Appendix. What is the time cost in the training period since it is involved with the large computation of ray-tracing. (This point is just a question but not weakness.)

[1] Caesar, Holger, et al. "nuscenes: A multimodal dataset for autonomous driving." Proceedings of the IEEE/CVF conference on computer vision and pattern recognition. 2020.


**Summary Of The Paper:**

This paper introduces object-LiDAR points to support weak supervised monocular 3D detection and obtain good results.

**Summary Of The Review:**

This paper proposes a method named WeakM3D to use object-LiDAR points to supervise the 3D learning from monocular images. In order to align the object-LiDAR points with monocular 3D predictions, the paper analyses the challenges and presents corresponding techniques or loss constraints. The method can train the models without manual labels, which is practical for monocular detection tasks.

---

### Official Review · Reviewer_1Zpn · 2021-11-01

**Correctness:** 3
**Technical Novelty And Significance:** 3
**Empirical Novelty And Significance:** 2
**Recommendation:** 6
**Confidence:** 4

**Main Review:**

** Strength **
- Clear motivation and good problem. Addressing the annotation problem for monocular 3D object detection is a task valuable in practice.

- The experiments are extensive and the results are encouraging. While the results are not outperforming all supervised methods, they do outperform previous weakly supervised methods and some previous supervised methods.

** Weakness **
- The proposed method does not estimate 3D box parameters fully. It relies on priors to estimate object sizes. As shown in the appendix, the object size is estimated by computing statistics from the annotated dataset.
In fact, in my opinion, to be fair, the results of learned dimensions should be shown in the paper, and the results of the frozen dimensions could be shown as a supplementary result.

- Section 2.1 mentions a few methods that are 'widely employed in autonomous driving'. Can this claim be supported by any evidence or citation? While LiDAR-based methods have higher accuracies, their adoption in the autonomous driving industry was not well documented.

- To avoid confusion, please indicate at the table caption that the experiments are for Car class. E.g., in Table 2, 3 this is not mentioned.

** Questions **

1) Both geometric alignment loss and ray tracing loss assume that if we take the bounding box of the observed LiDAR points, it should somewhat have similar sizes to the ground truth box if the annotation is done.
How do these two losses perform in the case of amodal detection where let's say, only the back of a car is observed? How well is the alignment work?

2) LiDAR point clouds can be noisy, i.e., the points observed across frames are inconsistent. How does this affect the weak supervision?

3) The weak supervision relies on preprocessing such as ground plane estimation and clustering, which imposes some limitations.
In my understanding, if clustering works well in the beginning, we might not need detection anymore. If clustering does not work, the weak supervision could be meaningless. Could you elaborate on which cases such preprocessing fails?
Does it make sense to regress boxes from the point clustering and compare to the ground truth boxes to see how well the clustering is?

4) The proposed method outperforms some previous methods in KITTI validation set, but does not outperform on KITTI test set, e.g., see RTM3D method on the KITTI leaderboard. And this is not shown in Table 2. Why is there such a performance gap?

5) The metrics used for experiments are rather inconsistent. I suggest reporting both AP11 and AP40 when suitable, and keeping all IoU to 0.7. I am confused about the switch from AP11 IoU 0.7 (Table 1) to AP40 IoU 0.7 (Table 2), and then to AP40 IoU 0.5 in Table 4 and the ablation study.

6) An inherent limitation of the method is the handling of occlusion. Due to viewpoint differences, some objects might appear in the LiDAR view but not in the image view and vice versa. How did the authors handle this problem?

7) Did the authors evaluate with nuScenes dataset? As far as I know, more papers report their performance on this dataset than Waymo. Also, this dataset also comes with evaluation metrics for translation, orientation, separately, which could provide more insights.

**Summary Of The Paper:**

The paper presents a new method for weakly supervised monocular 3D object detection. The authors are motivated by the fact that 3D bounding boxes for monocular video are expensive to annotate, and so propose to use 2D boxes and RoI on LiDAR point clouds to supervise the detector by minimizing the alignment between the RoI on LiDAR and the 3D box prediction. The results show that the proposed method achieves good performance on the KITTI and Waymo dataset.

**Summary Of The Review:**

The paper presents interesting ideas for training monocular 3D object detectors with just LiDAR point clouds supervision, and the proposed solution is reasonable and achieves decent performance. Therefore I am leaning a little positive despite there are several concerns that I invite the authors to address in the rebuttal.

---

### Official Review · Reviewer_Pqqk · 2021-11-02

**Correctness:** 3
**Technical Novelty And Significance:** 3
**Empirical Novelty And Significance:** 3
**Recommendation:** 8
**Confidence:** 4

**Main Review:**

This review points out the strengths and weaknesses of the paper in descending order. I have gone through the methods, technical implementation, and experiments in detail. In my opinion, the proposed method is sound under the specific setting of this paper.

**Strengths:**

I like the idea of this work mainly from two aspects:
1) This work proposes an idea that is straightforward but (seems) has not been explored in the community: Lidar points can provide weak supervision to estimate 3D object bounding boxes. The nature of Lidar makes the captured points close to object edges, which could briefly describe the shape. Meanwhile, the sparsity of point clouds largely reduces the cost to recognize background points. These two properties make this work possible.

2) This work notices the problem that Lidar points usually only exist in one edge since it captures scenes in one observation. It proposes a ray tracing loss to mimic the occlusion problem and encourage the network to predict 3D boxes that put ‘object-LiDAR-points’ to the edge toward the observer. This regularization term does make sense and effectively boost the detection performance. But by the way, ‘semantic ambiguity’ seems not to be an appropriate term to describe this problem, and is actually a bit distracting. It may lead people to think about object categories or something close.

The design of other components, such as geometry loss, also provides a complete training framework.

**Weaknesses:**

Major:
1) The derivation of geometry loss (Eq 1) and ray-tracing loss (Eq 2) relies on building virtual rays from a camera point. It assumes that the distance between the camera and Lidar and their observation angle difference are small enough to ignore, which is not always true in the real world. Rectification will not solve this problem because the observed planes may be different. I was hoping the authors could provide further discussion on this point, e.g., how could their method work in this setting or why the potential drawback is negligible.

2) It is good that the authors have noticed that Lidar points often gather along edges, and proposed a ray-tracing loss. However, this work did not explain or try to analyze how the network could predict a complete 3D bounding box, when only one object edge is constrained. For example, how would the network predict the length of a bounding box if all the Lidar points stay at a corner? In my humble opinion, the network may highly rely on the templates (or say priors) of height, width, and length (Sec 4.1). This might be supported by the observation that the proposed method performs well on easy metrics while has a clear gap against other methods on stricter metrics like AP moderate or hard, as shown in Table 1.

Minor：

3) The table captions are confusing, especially those on Page 8 and 9. For example, in Table 6, the caption is just ‘Comparisons on different requirements’. Table 3, ‘Comparisons on CenterNet’. Nothing meaningful is provided in such a short sentence, and the authors did not provide further explanations. Moreover, in Table 6, the Requirements are ‘2D box + LiDAR’ or ‘2D mask + LiDAR’. But in my understanding and the content of other sections, the proposed method does not need Lidar during inference. Although I could understand the meaning, the authors should provide comprehensive descriptions in the Section Experiment.


**Summary Of The Paper:**

This work, named WeakM3D, notices that Lidar points can provide weak supervision for monocular 3D Object Detection, dispensing with expensive 3D box annotations. It further identifies some challenges when trying to use Lidar points as supervision, and proposes several losses to mitigate the problems. Quantitative experiments on various datasets show that the method could match some fully supervised 3D object detection methods. I believe this paper is novel to some extent and has merits, though some of its claims need further support.

**Summary Of The Review:**

Overall I like some ideas of this work but it seems the authors should provide more analysis to support their claim. More discussion on how the proposed supervision works is also beneficial.

Overall I think this paper is a good starting point. I may raise the score if the authors could solve the concerns on loss derivation (weaknesses 1) and provide analysis on complete bounding box prediction (weaknesses 2). I guess weaknesses 2 cannot be fully avoided, but a detailed analysis can back up the authors’ claims much better.

---

### Official Review · Reviewer_fzTD · 2021-11-06

**Correctness:** 3
**Technical Novelty And Significance:** 3
**Empirical Novelty And Significance:** 2
**Recommendation:** 6
**Confidence:** 4

**Main Review:**

Strengths:
+ Novelty of addressing the weakly supervision: two introduced losses are reasonable for weak supervision from raw point clouds.
+ Organization of the paper is clear and easy to follow.

Weaknesses:
- The biggest concern is the motivation of the problem statement. When the LiDAR point clouds are available, why not use 3D point cloud based object detection as the supervision for the monocular object detection task? I suggest the authors show using the proposed method can achieve better performance than using the SOTA 3D object detection results as supervision.
- Reference Meng et al. (2020) is an important related work, which is also mentioned in the manuscript. But seems the authors are not doing any comparisons between their work and this reference. Any reasons?
- The paper names the backbone network as a 3D feature encoder, but seems the backbone is just a classical 2D image backbone without any 3D mechanisms (e.g., depth-aware convolution, etc.). I would suggest the authors try some more advanced backbones designed for 3D object detection, which may further improve the performance.


**Summary Of The Paper:**

This paper proposes a weakly supervised monocular 3D object detection framework using LiDAR point cloud as the supervision instead of 3D bounding boxes. In order to achieve this, the authors introduce geometric alignment loss and ray tracing loss to address point cloud alignment, loss balancing to address point cloud density issues, and a training strategy for learning disentanglement. The paper is overall well-written and clearly stated. However, the motivation of the problem is questionable and is not clear to me. Please see the detailed comments below.

**Summary Of The Review:**

Overall, this paper is in good shape. The most critical concern is motivation. If the authors can provide a reasonable explanation on the motivation as well as other concerns mentioned above, I would like to consider recommending acceptance.

---

### Decision · Program_Chairs · 2022-01-20

**Decision:**

Accept (Poster)

**Comment:**

This paper received 5 quality reviews. The rebuttal and discussion were effective and addressed many concerns from the reviewers, after which most reviewers increase their ratings of this paper. The final rating is 6 from 4 reviewers, and 8 from 1 reviewer. The AC concurs with the positive recommendation from the reviewers and recommends acceptance.